# Rate of Hospitalizations and Mortality of Respiratory Syncytial Virus Infection Compared to Influenza in Older People: A Systematic Review and Meta-Analysis

**DOI:** 10.3390/vaccines10122092

**Published:** 2022-12-07

**Authors:** Stefania Maggi, Nicola Veronese, Marianna Burgio, Giorgia Cammarata, Maria Elena Ciuppa, Stefano Ciriminna, Francesco Di Gennaro, Lee Smith, Mike Trott, Ligia J. Dominguez, Giovanni M. Giammanco, Simona De Grazia, Claudio Costantino, Francesco Vitale, Mario Barbagallo

**Affiliations:** 1Consiglio Nazionale delle Ricerche, Neuroscience Institute, 35100 Padova, Italy; 2Department of Health Promotion, Mother and Child Care, Internal Medicine and Medical Specialties “G. D’Alessandro”, University of Palermo, 90127 Palermo, Italy; 3Department of Biomedical Sciences and Human Oncology, Clinic of Infectious Diseases, University of Bari “Aldo Moro”, 70121 Bari, Italy; 4Centre for Health Performance and Wellbeing, Anglia Ruskin University, Cambridge CB1 1PT, UK; 5Centre for Public Health, Queen’s University Belfast, Belfast BT12 6BA, UK; 6School of Medicine and Surgery, University of Enna “Kore”, 94100 Enna, Italy

**Keywords:** Respiratory Syncytial Virus, influenza, meta-analysis, hospitalization, mortality

## Abstract

Respiratory Syncytial Virus (RSV) is commonly regarded as an infection typical of children, but increasing literature is showing its importance in older people. Since the data regarding the impact of RSV are still limited for older people, the aim of this systematic review and meta-analysis is to compare the rate of hospitalization and mortality between RSV and influenza in this population. A systematic literature search until 15 June 2022 was done across several databases and including studies reporting incidence rate and cumulative incidence of hospitalization and mortality in RSV and influenza affecting older people. Among 2295 records initially screened, 16 studies including 762,084 older participants were included. Compared to older patients having influenza, patients with RSV did not show any significant different risk in hospitalization (either cumulative or incidence rate). Similar results were evident for mortality. The quality of the studies was in general good. In conclusion, our systematic review and meta-analysis showed that the rate of hospitalization and mortality was similar between RSV and influenza in older adults, suggesting the importance of vaccination for RSV in older people for preventing negative outcomes, such as mortality and hospitalization.

## 1. Introduction

Respiratory Syncytial Virus (RSV) is commonly regarded as an infection typical of children [1,2]. Unlike influenza, for which the epidemiological importance has been widely recognized for years, the epidemiological impact of RSV infection in middle-aged and older adults has only recently gained importance [3,4]. In children, primary RSV infections may result in severe disease and subsequent infections that are often comparatively mild [1]. Sometimes, a state of incomplete immunity has been reported, that can predispose to a continued susceptibility to reinfection during the life [5].

Such as for influenza, older adults may have a greater incidence of negative outcomes when affected by RSV, since they are affected more frequently from respiratory and cardiovascular conditions [6], such as chronic obstructive pulmonary disease [7] or heart failure [8], which can further increase hospitalization and mortality rates and other negative outcomes [9]. In this regard, it was reported in the USA that each year, between 60,000–120,000 older adults are hospitalized and 6000–10,000 of them die due to RSV infection [10,11]. In particular, RSV is an important risk factor for hospitalization in frailer subjects, such as those living in nursing homes [12]. In particular, some systematic reviews with meta-analyses have reported that RSV could be associated with a high case fatality rate in nursing home [13,14].

Because of these findings, and the known role of influenza in causing hospitalization and mortality, studies of RSV have often used influenza as comparison group, also considering that it is often hard to distinguish between symptoms of severe RSV or severe influenza. In older people, it was reported that RSV infections present less frequently with fever but more frequently with wheezing [15], for example, but otherwise it can be difficult to distinguish it clinically from influenza without appropriate diagnostic tests.

Since the data regarding the impact of RSV are still limited for older people, the aim of this systematic review and meta-analysis is to compare the rate of hospitalization and mortality between RSV and influenza in older people including observational studies available.

## 2. Materials and Methods

This systematic review adhered to the PRISMA statement (checklist available in the Appendix A) [16]. The protocol was registered on 2 December 2022 on PROSPERO (CRD42018381040).

### 2.1. Data Sources and Searches

Four investigators (MEC, GC, SC, and MB) by couples, independently, conducted a literature search using several databases including Pubmed, Embase and Web of Science from database inception until 15 June 2022, including observational studies investigating the rate of hospitalization and/or mortality in older people affected by RSV infection.

The search strategy run in Pubmed was: “(Respiratory Syncytial Virus OR RSV OR Respiratory Syncytial Viruses [mh]) AND (“Mortality”[Mesh] OR “mortality” [Subheading] OR “Mortality” OR “Mortalities” OR “Case Fatality Rate” OR “Case Fatality Rates” OR “Death Rate” OR “Death Rates” OR “survival” OR “Hospital” OR “Acute care” OR “Hospitalized older adults” OR “Hospital medicine” OR “Older inpatients” OR “Inpatient” OR “Acute geriatric ward” OR “Clinical ward” OR “Medical units” OR “Geriatric units” OR “Hospitalization” OR “Length of Stay”) AND (((“elderly population”[Title/Abstract] OR elderly[Title/Abstract] OR elders[Title/Abstract] OR “elderly patient”[Title/Abstract] OR “elderly patients”[Title/Abstract] OR “elderly people”[Title/Abstract] OR “aged 70”[Title/Abstract] OR “aged 80”[Title/Abstract] OR “aged >65”[Title/Abstract] OR “>65 years old”[Title/Abstract]) OR (“70 years old”[Title/Abstract] OR “>70 years old”[Title/Abstract] OR “older person*”[Title/Abstract] OR “older persons”[Title/Abstract] OR “Older Adult*”[Title/Abstract] OR “Older Adults”[Title/Abstract] OR “Oldest Old”[Title/Abstract] OR Nonagenarians[Title/Abstract] OR Octogenarians[Title/Abstract] OR Centenarians[Title/Abstract])) OR (“aged over 80”[Title] OR “aged over 90”[Title]))” Any inconsistencies were resolved by consensus with a senior author (NV).

The search strategy was adapted using OVID for the other databases searched.

### 2.2. Study Selection

Following the PICOS question, we considered eligible studies that included older participants (defined as age ≥60 years) (P), affected by RSV (I) versus influenza or healthy people (C) including the incidence of hospitalization and/or mortality as outcomes (O). Therefore, prospective and retrospective studies were considered (S). We also included conference abstracts, if sufficient data were available for the meta-analysis. We excluded studies made in people younger than 60 years, considering outcomes (such as disability or intensive care unit admission) other than those mentioned above, and cross-sectional studies. Moreover, studies were excluded if data could not be meta-analyzable.

### 2.3. Data Extraction

Four independent investigators (MEC, GC, SC, and MB) extracted key data from the included articles in a standardized Excel spread sheet and a third independent investigator (NV) checked the data. For each article, we extracted data on author names, year of publication, country/continent, study design (retrospective or prospective), demographic information, and follow-up duration (in months).

### 2.4. Outcomes

The primary outcomes were the incidence rates of hospitalization and mortality. The incidence could be reported as overall cumulative incidence or as standardized incidence rates with their 95% confidence intervals (CIs). In this latter case, we used the formulas reported in the Cochrane handbook of systematic reviews of interventions [17]. All the data were reported as per 100,000 persons-year for hospitalization and per 1000 persons-year for mortality.

### 2.5. Quality Assessment

The Newcastle-Ottawa Scale (NOS) was used to assess the study quality/risk of bias [18]. The NOS assigns a maximum of nine points based on three quality parameters: selection, comparability, and outcome. The evaluation was made by four independent investigators (MEC, GC, SC, and MB) and checked by another (NV), independently. The risk of bias was then categorized as high (<5/9 points), moderate (6–7), or low (8–9) [19].

### 2.6. Data Synthesis and Analysis

All analyses were performed using STATA version 14.0 (StataCorp, College Station, TX, USA).

The primary analysis investigated the incidence rates of hospitalization/mortality between patients affected by RSV and influenza. We calculated the risk ratios (RRs) with their 95% confidence intervals (CIs), applying a random-effect model [20]. Similarly, we considered the mean difference (MD) with 95% CI in the incidence rate RSV and influenza standardized as indicated before. The results were reported by the method used for the diagnosis of RSV/influenza.

Heterogeneity across studies was assessed by the I^2^ metric and χ^2^ statistics and significant heterogeneity was placed in case of an I^2^ ≥ 50% or the correspondent *p*-value < 0.05.

Publication bias was assessed by visually inspecting funnel plots and using the Egger’s bias test [21]. The trim-and-fill analysis was planned for addressing this issue, but not used [22].

For all analyses, a *p*-value less than 0.05 was considered statistically significant.

## 3. Results

### 3.1. Search Results

As shown in Figure 1, among 2295 records initially screened, 74 were retrieved as full texts. After excluding some works, mainly because data were not meta-analyzable (Appendix A reports the references and the reason of the exclusion), 16 papers were finally included [23,24,25,26,27,28,29,30,31,32,33,34,35,36,37,38].

### 3.2. Study and Patient Characteristics

Full details regarding the descriptive findings are reported in Table 1. The 16 studies included 762,084 older participants (range: 29 to 551,633 participants), with a median for each study of 1246 older people. The mean age was 73.4 years. The majority of studies were conducted in North America (*n* = 9), five in Europe, one in Africa and another one in twelve different countries. No study was made in Asia. Overall, nine studies had a prospective design, and the remaining seven retrospective. The diagnosis of RSV/influenza was mainly made using polymerase chain reaction (PCR) (*n* = 9), sometimes in combination with other methods. The median follow-up time was 48 months, with a range between 3 and 84 months.

### 3.3. Rate of Hospitalizations and Mortality in RSV vs. Influenza: Meta-Analysis

As reported in Table 2, compared to patients having influenza, patients with RSV did not show any significant different risk in hospitalization. When considering cumulative incidence, the RR was 0.93 (95% CI: 0.53–1.62; *p* = 0.80; I^2^ = 0%; 5 studies) (Appendix A). The Egger’s test suggests the absence of any publication bias. Similarly, when including ten studies reporting data as incidence rate per 100,000 persons-year, we did not observe any difference in hospitalization rate (MD = −262; 95% CI: −755; 229), even if this outcome was characterized by a high heterogeneity (I^2^ = 99%), mainly driven by the study of Goncalo [30] that shows a reduced risk of hospitalization in RSV compared to influenza (Appendix A). No publication bias was evident.

When considering mortality as outcome, the results remained similar. Compared to influenza, RSV did not carry any significant increased risk in mortality (RR = 1.19; 95% CI: 0.98–1.45; *p* = 0.08; I^2^ = 0%; 4 studies) (Appendix A). No publication bias was present. Finally, when using incidence rate, instead of cumulative incidence, RSV did not differ in terms of mortality compared to influenza (MD = 15 per 1000 persons-year; 95% CI: −133 to 162; *p* = 0.85; I^2^ = 0%) (Appendix A). Again, no evidence of publication bias was present (funnel plots reported in Appendix A.

### 3.4. Sensitivity Analyses

We have categorized the incidence of mortality and hospitalization according to the diagnostic method of RSV. Regarding the cumulative incidence of death, all the studies included used the PCR; when using the incidence rate, no significant differences between studies using the weekly influenza update and antigenic methods were found (*p* for interaction = 0.613). Regarding the cumulative incidence of hospitalization, no significant differences between PCR, administrative data and luminex RVP FAST assay (*p* for interaction = 0.706), while the mean difference between RSV and influenza reported that the incidence rate collected using administrative data (weekly influenza update, ICD-10, HMDB-14) was statistically lower than PCR and antigen tests (*p* for interaction < 0.0001).

### 3.5. Risk of Bias

The Newcastle-Ottawa Scale overall indicated a good quality of the studies included, without any study at high risk of bias. The most common sources of bias were the ascertainment of exposure (that was not accurate in five studies and the comparability of cohorts on the basis of the design or analysis that was not of high quality since the propensity score was missing in all the studies included (Table 1 and Appendix A).

## 4. Discussion

In this systematic review with meta-analysis including 16 observational studies and a total of 762,084 older participants, we found that the incidence of hospitalizations and mortality was similar between RSV and influenza, overall reinforcing the importance of identifying RSV in older people.

In previous works, an important effort was dedicated to comparing the clinical severity and presentation of RSV with influenza [8,25,27,38]. In agreement with the previous studies, also included in this systematic review, we found that the hospitalization and the mortality rate attributable to RSV in older people was like that of influenza. Regarding the possible differences between RSV and influenza infections, one study reported that older individuals with RSV were admitted later in their illness, on average, than individuals with influenza. These findings may indicate a slower overall progression to acute illness for RSV versus influenza, as confirmed in other studies [39]. Similarly, another important work reported that the diagnosis of RSV infection in hospitalized older adults is often delayed and this may further affect clinical management and outcomes [40]. All these findings, in our opinion, indicate the urgent need for a stricter surveillance and also using molecular approaches for individuating RSV [41].

In this regard, it is important to remember that influenza represents a major cause of morbidity and mortality in older people [42]. Moreover, influenza is an important cause of other medical conditions in older subjects such as stroke and cardiovascular diseases [43], disability [44], pneumonia [45], and finally mortality. Vaccination against influenza in older people is highly effective in preventing all these conditions and therefore highly recommended [46]. Therefore, to report that the incidence rate of hospitalization and mortality is similar between RSV and influenza is of epidemiological importance, also considering the likely increased susceptibility observed during COVID-19 pandemic [47].

Based on the epidemiological findings confirmed by our systematic review with meta-analysis, current vaccine development efforts have identified prevention of severe RSV-associated illness. To the best of our knowledge, 33 respiratory syncytial virus prevention candidates are in clinical development using different approaches, including recombinant vector, subunit, particle-based, live attenuated, chimeric, and nucleic acid vaccines, and monoclonal antibodies [48,49]. Nine candidates are in phase 3 clinical trials [48]. Of interest, a phase 3 trial reported that a single dose of an adjuvant vaccine was highly efficacious against RSV-confirmed lower respiratory tract diseases and RSV-confirmed acute respiratory infections in older adults, regardless of RSV disease severity, RSV subtype, baseline comorbidity and presence of frailty [50].

The findings of this systematic review must be interpreted within its limitations. First, some outcomes were characterized by a high heterogeneity. Second, about half of the studies had a retrospective design, possibly introducing a bias in these findings. Third, no study was made in Asia, limiting the generalization of our findings in these countries. Finally, the admission rate in ICU was not available for almost all studies included in our systematic review, but the admission in ICU represents an important risk factor for mortality, as shown in pediatric population [51]. However, it has been reported that ICU admission rate is similar for RSV and influenza older patients [25].

## 5. Conclusions

Our systematic review and meta-analysis showed that the rate of hospitalization and mortality was similar between RSV and influenza in older adults, suggesting the importance of vaccination for RSV in older people for preventing the negative outcomes, such as hospitalization and mortality. Since RSV is highly prevalent in older people and associated with negative outcomes, our systematic review supports the need of increasing awareness before vaccines availability. Future studies confirming our findings in the light of public health interventions are needed.

## Figures and Tables

**Figure 1 vaccines-10-02092-f001:**
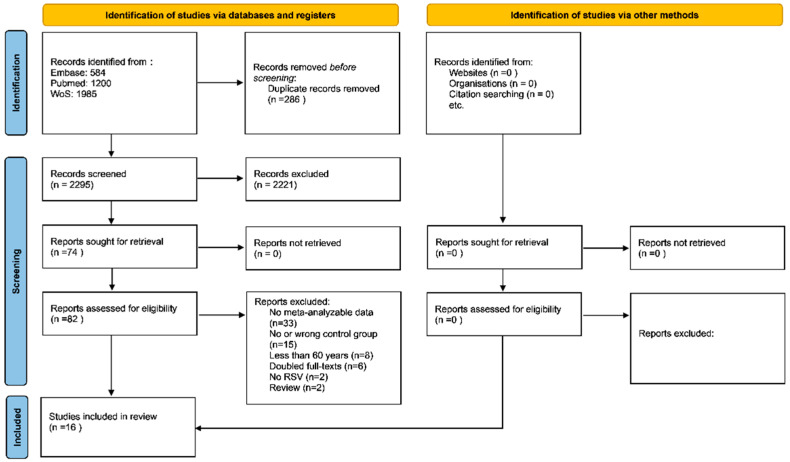
PRISMA flow-chart. Legend: n = number.

**Table 1 vaccines-10-02092-t001:** Descriptive characteristics.

Author, Year	Type of Study	Sample Size	Mean Age	Diagnosis of RSV and Influenza	Follow-Up (Months)	NOS
Ackerson, 2019	Retrospective	2523	78	PCR and culture	12	8
Auvinen, 2021	Prospective	974	76	PCR	48	8
Ellis, 2003	Retrospective	10,581	65+	Antigent tests and cultures	48	7
Falsey, 2005	Prospective	146	72	RT-PCR, Serologic test, viral culture	48	8
Falsey, 2021	Prospective	604	65.6	PCR	3	8
Gilca, 2014	Prospective	210		Luminex RVP FAST assay	48	7
Gonçalo Matias, 2017	Retrospective	64,456	65+	Weekly influenza update	144	7
Korsten, 2020	Prospective	1040	75	PCR	48	8
Loubet, 2016	Prospective	1452	74	PCR	12	8
Malosh, 2017	Prospective	426		PCR	24	8
Muller-Pebody, 2006	Prospective	551,633	65+	ICD-10	36	7
Rabarison, 2019	Retrospective	375		PCR	60	8
Schanzer, 2008	Retrospecitve	103,262		Hospitalization Morbidity Database (HMDB)14	60	7
Sharp, 2021	Retrospective	21,787	65–74	Antigene detection, culture, and genomic/pcr/lcr detection	84	8
Tseng, 2017	Prospective	2586	60+	PCR	48	8
Widemer, 2012	Prospective	29	65+	PCR	36	8
Total	9 studies: prospective; 7 studies: retrospective	762,084	73.4	9 studies: PCR; 7 studies: others	48 (range: 3–84)	

**Table 2 vaccines-10-02092-t002:** Meta-analysis of hospitalization and mortality rate between RSV and influenza.

	Cumulative Incidence						Incidence Rate (Per 100,000 Persons-Year)					
Outcome	N of Studies (Participants)	RR	95% CI	*p*-Value	I^2^	Egger’s Test (*p*-Value)	N of Studies	MD	95% CI	*p*-Value	I^2^	Egger’s Test (*p*-Value)
Hospitalization	5	0.93	0.53–1.62	0.80	0	−0.73 (*p* = 0.08)	10	−262	−755; 229	0.30	99	−6.57 (*p* = 0.20)
Mortality	4	1.19	0.98–1.45	0.08	0	0.57 (*p* = 0.10)	2	15	−133; 162	0.85	0	Not possible

Legend: RR: risk ratio; CI: confidence intervals; MD: mean difference.

## Data Availability

The data are available upon reasonable request to the corresponding author.

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
