# Peer review of "Rate of Hospitalizations and Mortality of Respiratory Syncytial Virus Infection Compared to Influenza in Older People: A Systematic Review and Meta-Analysis"

_vaccines, 2022, doi:10.3390/vaccines10122092_

Round 1

Reviewer 1 Report

This review aims to compare the rate of hospitalization and mortality between Respiratory Syncytial Virus (RSV) and influenza in older people.

Based on this review, the authors concluded that the rate of hospitalization and mortality was similar between RSV and influenza in older adults, suggesting the importance of vaccination for RSV in older people for preventing negative outcomes, such as mortality and hospitalization.

The introduction section is functional to the review's aims.

In the material and methods section, I strongly encourage the authors to record the review in PROPERO database, considering that they have utilized the PICO's model.

The results summarized the main data of their meta-analysis.

The discussion section should be improved. Despite I appreciate the choice of the authors to include the section limitations, several improvements should be made. Particularly, the authors should improve this section, focusing on the international data about the main data obtained through revision.

Finally, the conclusion section should be improved, highlighting the take-home message, the future implications, and future research lines.

Author Response

Reviewer 1

In the material and methods section, I strongly encourage the authors to record the review in PROPERO database, considering that they have utilized the PICO's model.

R: We sincerely thank the Reviewer for this comment. We have now recorded the protocol in PROSPERO, as follows:

“The protocol was registered on 02nd December 2022 on PROSPERO (CRD42018381040).”  

The discussion section should be improved. Despite I appreciate the choice of the authors to include the section limitations, several improvements should be made. Particularly, the authors should improve this section, focusing on the international data about the main data obtained through revision.

R: Good point. We thank the Reviewer for this comment. We have now updated the Limitations section, adding considerations of the international panorama, as suggested.

Finally, the conclusion section should be improved, highlighting the take-home message, the future implications, and future research lines.

R: Good point. We have now added this sentence in order to improve the conclusion section, as follows:

“Since RSV is highly prevalent in older people and associated with negative outcomes, our systematic review supports the need of increasing awareness before vaccines availability. Future studies confirming our findings in the light of public health interventions are needed.”

Reviewer 2 Report

Estimated Authors,

I've read with great interest the present systematic review on the occurrence of RSV infections in older individuals, and focusing on two main outcomes represented by hospitalizations and deaths. Briefly, Authors were able to rule out any substantial excess of risk for both outcomes.

The articles is well written, and well organized. I've only a couple of suggestions, and more precisely:

1) as some previous studies have suggested a substantial increase in mortality rates in nursing homes (e.g.  Shi et al. Journal of Infectious Diseases 2020;222:S563-569; Shi et al. Journal of Infectious Diseases 2020;222:S577-583), Authors should provide some insights on the settings of the studies they retrieved (e.g. community vs. hospital vs. nursing homes);

2) Authors should provide information about the share of patients that were diagnosed in settings such as UCI: for example in a recent study on febrile seizures in RSV cases (https://pubmed.ncbi.nlm.nih.gov/36412662/), being the patient diagnosed in pediatric ICU represented a substantial risk factor for the eventual outcome, as the patient was by design in more severe condition at the beginning of the study: could your estimated having been similarly affected? By providing subgroup estimates you could rule out this option [and, please: I've only cited this study as an example, don't feel suggested or recommended to cite it].

3) Authors should report their estimates for hospitalization and deaths by diagnostic procedure (i.e. antigenic vs. PCR) for reasons quite similar to those reported in point 2.

4) please include even as supplementary files the radial and funnel plots.

Author Response

Reviewer 2

The articles is well written, and well organized.

R: We thank the Reviewer 2 for his/her comments that together with the comments of the other Reviewer have further improved our systematic review.

I've only a couple of suggestions, and more precisely:

1) as some previous studies have suggested a substantial increase in mortality rates in nursing homes (e.g.  Shi et al. Journal of Infectious Diseases 2020;222:S563-569; Shi et al. Journal of Infectious Diseases 2020;222:S577-583), Authors should provide some insights on the settings of the studies they retrieved (e.g. community vs. hospital vs. nursing homes)

R: We sincerely thank the Reviewer for this important comment. We have now added in the Introduction section this sentence, as follows:

“In particular, some systematic reviews with meta-analyses have reported that RSV could be associated to a high case fatality rate in nursing home.[13,14]

2) Authors should provide information about the share of patients that were diagnosed in settings such as UCI: for example in a recent study on febrile seizures in RSV cases (https://pubmed.ncbi.nlm.nih.gov/36412662/), being the patient diagnosed in pediatric ICU represented a substantial risk factor for the eventual outcome, as the patient was by design in more severe condition at the beginning of the study: could your estimated having been similarly affected? By providing subgroup estimates you could rule out this option [and, please: I've only cited this study as an example, don't feel suggested or recommended to cite it].

R: Good point. We acknowledge that the admission in ICU is an important outcome for older people affected by RSV. Unfortunately, this outcome was beyond the aims of our systematic review. Therefore, we have added this as potential limitation of our study, as follows:

“Finally, the admission rate in ICU was not available for almost all studies included in our systematic review, but the admission in ICU represents an important risk factor for mortality, as shown in pediatric population. [52] However, it has been reported that ICU admission rate is similar for RSV and influenza older patients. [25]”

3) Authors should report their estimates for hospitalization and deaths by diagnostic procedure (i.e. antigenic vs. PCR) for reasons quite similar to those reported in point 2.

R: We thank the Reviewer for this question. We have now added this paragraph in the Results section, as suggested:

“3.4 Sensitivity analyses

We have categorized the incidence of mortality and hospitalization according to the diagnostic method of RSV. Regarding the cumulative incidence of death, all the studies included used the PCR; when using the incidence rate, no significant differences between studies using the Weekly influenza update and antigenic methods were found (p for interaction=0.613). Regarding the cumulative incidence of hospitalization, no significant differences between PCR, administrative data and luminex RVP FAST assay (p for interaction= 0.706), while the mean difference between RSV and influenza reported that the incidence rate collected using administrative data (weekly influenza update, ICD-10, HMDB-14 ) was statistically lower than PCR and antigen tests (p for interaction < 0.0001).”

4) please include even as supplementary files the radial and funnel plots.

R: Done.